# AWA and ASH Homologous Sensing Genes of *Meloidogyne incognita* Contribute to the Tomato Infection Process

**DOI:** 10.3390/pathogens11111322

**Published:** 2022-11-10

**Authors:** Yuxin Li, Qiaona Ren, Tingting Bo, Minghe Mo, Yajun Liu

**Affiliations:** State Key Laboratory for Conservation and Utilization of Biological Resources in Yunnan Province, Kunming 650032, China

**Keywords:** *Meloidogyne incognita*, signal-sensing homologous genes, tomato roots, host perception

## Abstract

The AWA neurons of *Caenorhabditis elegans* mainly perceive volatile attractive odors, while the ASH neurons perceive pH, penetration, nociception, odor tropism, etc. The perceptual neurons of *Meloidogyne incognita* have been little studied. The number of infestations around and within tomato roots was significantly reduced after RNA interference for high-homology genes in AWA and ASH neurons compared between *M. incognita* and *C. elegans.* Through in situ hybridization, we further determined the expression and localization of the homologous genes *Mi-odr-10* and *Mi-gpa-6* in *M. incognita*. In this study, we found that *M. incognita* has neuronal sensing pathways similar to AWA and ASH perception of *C. elegans* for sensing chemical signals from tomato roots. Silencing the homologous genes in these pathways could affect the nematode perception and infestation of tomato root systems. The results contribute to elucidating the process of the plant host perception of *M. incognita*.

## 1. Introduction

Plant-parasitic nematodes are a class of seriously damaging pathogens that are exclusively parasitic in plants and have characteristics such as widespread parasitism, adaptability, and ease of transmission [1,2]. The second-stage juveniles (J2s) of the root- knot nematodes (RNKs) infect the roots of plants, followed by root-tissue protrusion to form root knots; the formation of root knots prevents the plant roots from absorbing and transporting water and nutrients and, due to nutrient deficiency, the plant grows short, declines in yield, withers, and dies [3,4]. Plant-parasitic nematodes cause an estimated USD 173 billion in agricultural losses annually [5].

Chemotaxis has long been considered the primary reason for nematode host localization. Nematode chemotaxis refers to the nematode′s use of head-sensing organs to sense water-soluble, volatile compounds produced by roots or inter-root microorganisms and to move and locate plant roots in response to changes in compound concentration [6,7]. Plant roots secrete many signaling substances, and RNKs can recognize plant hosts by sensing the signaling substances secreted by the plant root system [8]. Murungi et al. confirmed that the tomato root extracts methyl salicylate and tridecane were highly attractive to *Meloidogyne incognita* [9]. The root secretion of lauric acid of *Chrysanthemum coronarium* was able to attract *M. incognita* at low concentrations [10]. Studying the chemotaxis of plant-parasitic nematodes to their hosts and interfering with their localization to host roots is a new strategy for nematode management and control [11].

*Caenorhabditis elegans* has a highly developed chemosensory system that relies on its elaborate olfactory system to survive in the soil, sensing signals related to food, danger, or other animals [12]. There are 12 pairs of chemosensory neurons in the head of *C. elegans* that detect the external environment, and these are symmetrically distributed on both sides of the head [13,14]. Volatile chemicals that can be detected by AWA and AWC neurons mainly include alcohols, ketones, esters, amines, organic acids, aromatics and heterocycles [15,16]. AWA receptor ODR-10 leads to the perception of volatile compounds diacetyl [17]. AWA neurons also need ODR-7 to sense most of the compounds, and the loss of *odr-7* causes the impairment of AWA neurons′ olfactory sensation [18]. ASH neurons are mainly involved in the nematode avoidance response induced by high osmotic pressure, heavy metals, detergents, bitter alkaloids, acidic pH, and some organic smells [19]. Gene *gpa-11* is expressed on ASH neurons. It regulates the response to octanol by regulating serotonin, and jointly regulates the avoidance of quinine by *odr-3* and *gpa-3* [20]. PLC family genes *plc-1*, *egl-8*, *unc-13*, *elo-1* encode and regulate the phospholipase pathway [21]. The PLC pathway is activated to produce the second messenger inositol triphosphate (IP_3_) and diacetylglycerol (DAG), activate protein kinase (PKC), and release intracellular Ca^2+^ [22]. TPRV ion channels are expressed in the nerves of *C. elegans* and regulate olfactory response, mechanoreceptor response, and osmotic response of the nematodes. Gene *osm-9* and *ocr-2* encode cyclic adenonucleotide ion channels, which play a role in insect light transmission, vertebrate pain perception, non-neural pressure perception and osmotic pressure perception [23,24].

Compared to the rich chemosensory gene pool of *C. elegans*, the chemotaxis-related genes of root-knot nematodes are less studied. Lauric acid in root exudates of *Chrysanthemum coronarium* regulated the specific expression of *Mi-flp-18* gene encoding FMRFamide-like peptide neural regulator in a concentration-dependent manner [10]. Shivakumara et al. found that the movement track of *M. incognita* J2s was abnormal, and the number of nematodes that tended to infect the root tip of the tomato plant was significantly reduced after RNA interference the homologues *Mi-odr-1*, *Mi-odr-3*, *Mi-tax-2* and *Mi-tax-4*. In ISH analysis, *Mi-odr-1* mRNA seems to be located in the cell cluster related to amphibious neurons and phasmids [25]. In this study, we further aimed homologous genes of the AWA and ASH neurons in *M. incognita* and explored the signaling pathways of the southern root-knot nematode that are involved in the sensing of the tomato root system and root-infestation ability. It is helpful to understand the process of root-parasitic nematodes perceiving plant hosts and provide a new target for the control of parasitic nematode.

## 2. Results

### 2.1. Relative Expression of Homologous Gene on AWA and ASH Neurons after RNA Interference

After 20 h of RNA interference with homologous genes on AWA neurons, each homologous gene on AWA was repressed to different degrees. The relative expression of homologous genes *Mi-unc-13*, *Mi-elo-1*, *Mi-odr-10*, *Mi-egl-8*, *Mi-plc-1*, *Mi-osm-9*, *Mi-odr-7 Mi-gpa-6*, and *Mi-gpa-11* were significantly down-regulated on AWA neurons compared to the wild type and control template (Figure 1A).

After 20 h of RNA interference with homologous genes on ASH neurons, the relative expression of *Mi-gpa-11* and *Mi-osm-9* were significantly down-regulated compared to the wild type and control template (Figure 1B).

### 2.2. Motility of M. incognita after RNA Interference with the AWA and ASH Homologous Genes

There was no difference in the motility of *M. incognita* following RNA interference with the homologous genes *Mi-unc-13*, *Mi-elo-1*, *Mi-odr-10*, *Mi-egl-8*, *Mi-plc-1*, *Mi-osm-9, Mi-gpa-6,* and *Mi-odr-7* compared to the wild type and control template Figure 2B–D. There was no difference in motility of *M. incognita* after RNA interference with the homologous genes *Mi-gpa-11* and *Mi-osm-9* on ASH neurons compared to the wild type Figure 2E,F. This indicates that RNA interference with the related genes does not affect chemotaxis by impairing motility.

### 2.3. Chemotaxis of M. incognita to Tomato Root after RNA Interference with the AWA and ASH Neurons’ Homologous Genes

The receptor genes *Mi-odr-10* and *Mi-odr-7* were subjected to RNA interference, and the number of nematodes that came around the roots was significantly reduced (Figure 3A,B). After RNA interference for four genes of the PLC pathway *Mi-unc-13*, *Mi-elo-1*, *Mi-egl-8* and *Mi-plc-1* in AWA neurons in AWA, the number of nematodes aggregated around the roots was also significantly reduced (Figure 3A). In addition, RNA interference with the gene *Mi-gpa-6* and *Mi-osm-9* also reduced the number of nematodes around the roots (Figure 3C). These results suggest that there is a sensory pathway similar to that for AWA neurons in the process of the root perception of tomatoes by *M. incognita*.

After RNA interference with the G-protein gene *Mi-gpa-11* in ASH neurons, the number of nematodes around the roots was significantly reduced (Figure 3D). *Mi-osm-9* jointly encodes TPRV ion channels and is expressed in both AWA and ASH neurons. After the gene *Mi-osm-9* was subjected to RNA interference, the chemotaxis of nematode to root was significantly reduced (Figure 3D). These results suggest that there is a pathway similar to that involving ASH neurons in the process of tomato root perception by *M. incognita*.

### 2.4. Infectivity to Tomato Root of M. incognita after RNA Interference with AWA and ASH Neurons’ Homologous Genes

Like the chemotaxis of *M. incognita,* after receptor genes *Mi-odr-10* and *Mi-odr-7*, four PLC pathway genes *Mi-plc-1*, *Mi-egl-8*, *Mi-unc-13* and *Mi-elo-1*, and genes *Mi-gpa-6* and *Mi-osm-9* in AWA neurons were RNA interfered, nematode-infected roots were significantly reduced (Figure 4A–C). It is suggested that the AWA neurons′ sensing pathway was necessary for the infection of *M. incognita*.

*Mi-gpa-11* and *Mi-osm-9* in ASH neurons were subjected to RNA interference. After that, the number of infecting nematodes in roots was significantly reduced (Figure 4D). These results suggest that the sensing pathway of ASH neurons is involved in the infection of *M. incognita*.

### 2.5. In Situ Hybridization of Receptor Genes in Neurons of M. incognita

Two receptor genes, *Mi-gpa-6* and *Mi-odr-10*, in the AWA neurons of *M. incognita* were analyzed by in situ hybridization. Through hybridization, color rendering, and microscopic observation, it was observed that the genes *Mi-gpa-6* (Figure 5A) and *Mi-odr-10* (Figure 5B) had hybridization signals at the head neurons of *M. incognita*.

## 3. Discussion

After RNA interference with 9 homologues genes of *M. incognita* (*Mi-unc-13*, *Mi-elo-1*, *Mi-odr-10*, *Mi-egl-8*, *Mi-plc-1*, *Mi-odr-7*, *Mi-gpa-6*, *Mi-osm-9* and *Mi-gpa-11*), the perception and infection abilities of *M. incognita* for tomato root were significantly weakened.

Studies have shown that the chemical signal transduction of *C. elegans* depends on the G-protein-coupled receptor signal pathway. The cGMP signal pathway of the second messenger synthesized by guanylate cyclase is the cyclic nucleotide-gated CNG channel (TAX-2 and TAX-4), or the transient receptive potential TRPV channel (OSM-9 and OCR-2) mediated by polyunsaturated fatty acid PUFA [26,27]. Zhu et al. [28] studied how nematodes recognize attractive molecules indole and 2-ethylhexanol. The experimental results showed that the chemotaxis of indole and 2-ethylhexanol requires *str-193* on AWC and *str-7* encoded G-protein coupled receptors on AWA. In further genetic screening of downstream effectors of olfactory signal cascade, Gα subunit GSA-1, guanylyl cyclase ODR-1, DAF-11, and cGMP-gated channel TAX-2/TAX-4 are necessary for indole sensing, while the PLC pathway activated by TRPV channel OSM-9/OCR-2 and GPA-6 is responsible for detecting 2-ethylhexanol.

Shivakumara et al. [25] cloned four allelopathic genes (*Mi-odr-1*, *Mi-odr-3*, *Mi-tax-2* and *Mi-tax-4*) homologous to *C. elegans* in *M. incognita*. The movement track of the *M. incognita* J2s was abnormal, and the number of nematodes tending and invading tomato roots was significantly reduced after RNA interference with these genes. Moreover, the response to the low-dose nematode pheromone ascoroside # 18 was weakened. The results showed that *Mi-odr-1*, *Mi-odr-3*, *Mi-tax-2*, and *Mi-tax-4* played an important role in the perception of volatile and non-volatile compounds of *M. incognita.* Using qPCR analysis, the authors speculate that *Mi-tax-2* and *Mi-tax-4* may function downstream of *Mi-odr-1* and *Mi-odr-3* in the chemotaxis pathway of *M. incognita*. It is speculated that *M. incognita* perceives chemical gradients via one or more core chemosensory genes, such as *Mi-odr-1* and *Mi-odr-3,* followed by *Mi-tax-2* and *Mi-tax-4*, transducts signaling in the sensory organs, and selectively chemo-orients to specific cues. The gene *odr-1* is a gene on AWB and AWC neurons, encoding membrane-bound guanylyl cyclase (GCY) [29]. Gene *odr-3* regulates the production of cyclic nucleotides (cGMP) [30]. Gene *tax-2* and *tax-4* are involved in the activation of cyclic guanylate-gated channels [31].

Our study focused on allelopathic homologous genes similar to the genes in the AWA and ASH neurons of *C. elegans*. The *odr-10* gene encodes a receptor that senses volatile compounds and the attractive compound diacetyl, on AWA. The sensitivity of an *odr-10* mutant to the volatile attractant diacetyl was reduced by 100 times [28]. The receptor gene *odr-7* encodes an essential receptor for AWA neurons. Studies have proven that AWA neurons need *odr-7* to sense most compounds. Its role is to promote the differentiation of AWA neurons. The loss of *odr-7* caused olfactory impairment in AWA neurons [32]. After RNA interference with the homologous genes *Mi-odr-10*, *Mi-odr-7*, *Mi-plc-1*, *Mi-egl-8*, *Mi-gpa-6*, and *Mi-osm-9* in AWA neurons, it was found that the chemotaxis and infection of *M. incognita* for tomato roots decreased significantly. Based on the background of *C. elegans*, it is speculated that the signaling pathway involving the root-knot nematode’s AWA neurons for sensing tomato roots is as follows: the root signal binds to the receptor ODR-10; activates the G-protein GPA-6; activates the PLC pathway regulated by PLC-1/EGL-8 downstream; produces the second messengers diacylglycerol (DAG) and inositol triphosphate (IP_3_); activates the protein kinase PKC, resulting in the production of polyunsaturated fatty acids (PUFAs); regulates OSM-9; and then generates signal transduction that opens the TPRV ion channel. As there are many kinds of tomato root signals, our study only preliminarily explored the roles of the genes *Mi-odr-7* and *Mi-odr-10* in sensing tomato root signals, but it is not clear what specific types of compounds from tomato roots could be sensed, or which signals can be regulated.

The signaling pathway by which *C. elegans*’ ASH neurons sense the corresponding compounds is still imperfectly understood [33]. The corresponding receptor genes in ASH neurons have not been reported, and the genes that regulate the phospholipase pathway remain unknown, so the related signaling pathways of ASH neurons still need to be explored and improved. Upon RNA interference with homologous genes in ASH neurons, the chemotaxis and infectivity of *M. incognita* for tomato roots were observed to decrease significantly. It is speculated that the signaling pathway for the sensing of tomato roots by ASH-like neurons in *M. incognita* includes the following: the receptor activates the G-protein GPA-11, activates the phospholipase pathway to produce the second messenger, regulates OSM-9, and then opens the TPRV ion channel. 

Shivakumara et al.’s [25] in situ hybridization experiment of four homologous genes *Mi-odi-1*, *Mi-odr-3*, *Mi-tax-2*, and *Mi-tax-4* of the *M. incognita*, found that there were hybridization signals in the head and tail receptors of the *M. incognita*, indicating that the genes expressed in the head and tail of the nematode. Lans et al. [34] determined the location of the genes *gpa-6*, *gpa-5*, and *gpa-13* on *C. elegans* by an immunostaining localization method, and all of them expressed in the head. For two homologous receptor genes *Mi-odr-10* and *Mi-gpa-6*, we found that they both had hybridization signals at the head of *M. incognita*.

The allelopathic system of root parasitic nematodes is involved in many processes, such as migration, host recognition, host infection, and the establishment of permanent feeding sites in the host. At present, a variety of compounds related to the chemical perception of root parasitic nematodes have been reported, but the molecular mechanism of their allelopathy is less studied; in particular, the signal transduction pathway needs more in-depth research. This study confirmed that the neuronal pathways similar to those of the AWA and ASH of *C. elegans* are involved in the root-signal-sensing process of *M. incognita*. Based on the identification and location of root-knot nematodes at plant hosts, a new way to prevent root-knot nematodes from recognizing plants is suggested. Due to the complexity of the root system, the chemotaxis shown by *M. incognita* may be mediated by a variety of signaling substances. Further research is needed to determine the types of substances that are perceived by AWA and ASH, resulting in signal transduction.

## 4. Materials and Methods

### 4.1. Culture and Collection of M. incognita

A three-week-old tomato seedling (Bela, purchased from the Shandong Shengda seed industry) was transplanted to a pot (18 cm high and 8.5 cm in diameter). Sterilized humus, vermiculite and perlite (V:V:V, 7:2:10) were used as culture substrates. About 2000 J2s of *M. incognita* were inoculated on tomato roots [35]. The inoculated tomato seedlings were placed in greenhouse for about 50 days. The mature egg mass from tomato roots was picked and placed in a 200 µm mesh screen with sterile water. The *M. incognita* J2s that hatched in the incubator were filtered and collected in a 15 mL centrifuge tube and centrifuged at 4000 rpm for 5 min; the supernatant was discarded, and the nematode concentration was adjusted for analysis.

### 4.2. RNA Interference for Homologous Genes in AWA and ASH Neurons

BLAST protein comparison was performed for homologous genes related to the AWA and ASH neurons in *C. elegans* (https://wormbase.org//#012-34-5)(accessed on 12 November 2021) and *Meloidogyne* database (https://meloidogyne.inrae.fr/)(accessed on 12 November 2021) to select homologous genes with protein homology higher than 40%. See Appendix A for the comparison results of homologous genes between *Meloidogyne incognita* and *Caenorhabditis elegans*. See Appendix A for the comparison of protein homology between *Meloidogyne incognita* and *Caenorhabditis elegans*.

The conserved functional region of the candidate allelopathic homologous gene of *M. incognita* was searched in the NCBI database. According to the requirements of the in vitro transcription MEGA script RNAi Kit (AM1626, Thermo Fisher Scientific, Shanghai, China), the homologous gene primer was designed with Primer3, and the T7 promoter sequence was added at the 5′ end of the primer. The qPCR primers were designed according to the experimental requirements. *β-actin* was used as an internal reference gene as previously reported. See Appendix A for the primer information of RNAi in *Meloidogyne incognita*.

The total RNA of *M. incognita* J2s was extracted with the RaPure Total RNA Micro Kit (R4012, Magen). The cDNA was synthesized according to the instructions of Prime Script II 1st strand cDNA synthesis Kit (Takara). Plasmids were extracted with the E.Z.N.A. Plasmid DNA Mini Kit II (OMEGA) and sequenced by Tsingke Biotechnology Co., Ltd (Beijing, China). The MEGAscript RNAi Kit (AM1626, Thermo Fisher Scientific—the RNAi kit is used to synthesize a large number of dsRNA for RNAi experiments in nonmammalian systems. The Control Template is a linear dsDNA fragment with opposing T7 promoters. It yields a 500 bp dsRNA product) was used for in vitro transcription, and the dsRNA concentration was adjusted to 1 mg/mL.

A 1.5 mL RNase-free centrifuge tube was used to collect 200 µL (about 20,000 pieces) of newly-hatched *M. incognita* J2s; 10 µL of 5% m-dihydroxybenzene, 50 µL of spermidine (30 mM), 5 µL of 5% gelatin, and 100 µL of dsRNA were added, and M9 buffer (5.8 g of Na_2_HPO_4_·7H_2_O, 3.0 g of KH_2_PO_4_ and 5.0 g NaCl) was used to make up the volume to 500 µL. No dsRNA was added to the wild type, and dsRNA for genes was added to the gene-interference group. *M. incognita* J2s were immersed in the immersion system for 20 h, and then recovered with M9 buffer for 20 h. The in vitro immersion conditions are reported in the relevant literature [36]. Three replicates were established, and the test was repeated three times. 

The total RNA of *M. incognita* J2s treated in the experiment was extracted using the RaPure Total RNA Micro Kit (R4012, Magen). The RNA was reverse-transcribed into cDNA using the PrimeScript RT Reagent Kit with gDNA Eraser (RR047A, Takara). TB Green Premix Ex Taq (RR820A, TaKaRa) was used to detect the gene expression. The 2⁻^ΔΔ Ct^ method was used to calculate the relative gene expression.

### 4.3. Detection of Motility of M. incognita after RNA Interference

Tris-MES glue was prepared (23 g of Tris-MES and 100 mL of sterile water). After the glue was fully dissolved, 2.5 mL was aspirated and added to a 3 cm plate. The plate was divided into two areas, A and B (Figure 1A). About 200 RNAi-transfected *M. incognita* in zone B were stored in an incubator at 28 °C for 24 h. The numbers of nematodes in areas A and B were counted, and the mobility was calculated. Three replicates were established, and the test was repeated three times. Mobility (%) = (number of nematodes in area A/total number of nematodes in areas A and B) × 100%.

### 4.4. Detection of Chemotaxis and Infectivity of M. incognita for Tomato Roots after RNAi

Tomato seeds (selected cooperation 908, Taishu Seed Industry Co.,Ltd, Xi’an, China) were germinated to about 2 cm on filter paper for analysis. Tomato sprouts were added to the Pluronic F-127gel plate [37], and about 200 RNAi-transfected *M. incognita* J2s were added 1.5 cm away from the sprouts. The chemotaxis and infectivity of *M. incognita* for tomato roots after RNAi were observed under a microscope at 4, 8, 12, and 24 h after treatment. Three replicates were established, and the test was repeated three times. Then, the buds were soaked in 1% sodium hypochlorite for 30 min, washed with sterile water and boiled in acid fuchsin solution (3.5% acid fuchsin; 25% acetic acid) for 10 min, for counting the number of invaded nematodes.

### 4.5. In Situ Hybridization Experiment

The probes were labeled using the PCR DIG Probe Synthesis Kit (Roche) and amplified by two-step PCR. The first step was double-primer amplification. The PCR consisted of the following: 3 µL of 10 × PCR buffer, 5 µL of dNTPs, 1.5 µL each of the forward and reverse primers, 0.25 µL of the Ex Taq-enzyme, 2 µL of the cDNA template, and enough ddH_2_O to make up a total volume of 50 µL. The PCR conditions were 95 °C pre-denaturation for 4 min, 98 °C for 10 s, 52 °C for 20 s, and 72 °C for 1 min, for 35 cycles, followed by extension at 72 °C for 10 min. After the purification and recovery of the first-step amplification product as a template, the digoxin-labeled probe was obtained by single-primer amplification with the forward primer or reverse primer, respectively. The PCR solution was prepared as follows: 3 µL of 10× PCR buffer, 3 µL of the digoxin-labeled dNTP mix, 1.5 µL of the forward or reverse primer, 0.6 µL of Ex Taq, 2 µg of the DNA template, and enough ddH_2_O to make up the volume to 30 µL. The PCR conditions were as described above. The freshly hatched *M. incognita* J2s were fixed with paraformaldehyde at 4 °C for 24 h. The nematode was cut off with a surgical blade and then treated with protease K (1 mg/mL), methanol, and acetone precooled at −80 °C. Then, the nematodes were pre-hybridized by suspension in a hybridization solution. Finally, digoxin-labeled probes were added to hybridize at 40 °C for 16 h. After hybridization, the chromogenic solution was added to develop color overnight at 25 °C in the dark. Observations were made and photos were taken under a microscope (Nikon ECLIPSE, 80I, Nikon, Tokyo, Japan).

### 4.6. Data Processing

GraphPad Prism 8.0 was used to process all the experimental data. Two–way ANOVA and Tukey’s multiple comparisons were used for signification analysis.

## Figures and Tables

**Figure 1 pathogens-11-01322-f001:**
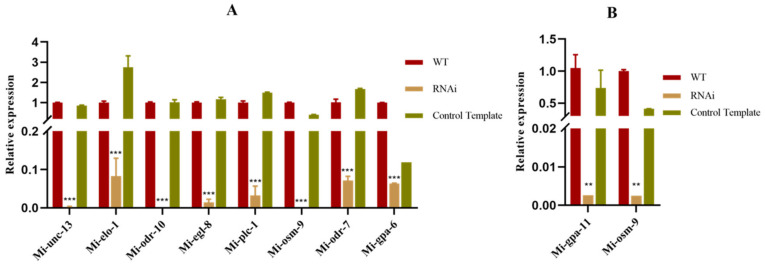
Expression of homologous genes in AWA and ASH neurons after RNAi: (**A**) the relative gene expression of AWA neuronal homologous gene after RNAi; (**B**) the relative gene expression of ASH neuronal homologous gene after RNAi. Each bar represents the standard error of the mean. Significant differences using Tukey’s multiple comparisons (** *p* < 0.01, *** *p* < 0.001).

**Figure 2 pathogens-11-01322-f002:**
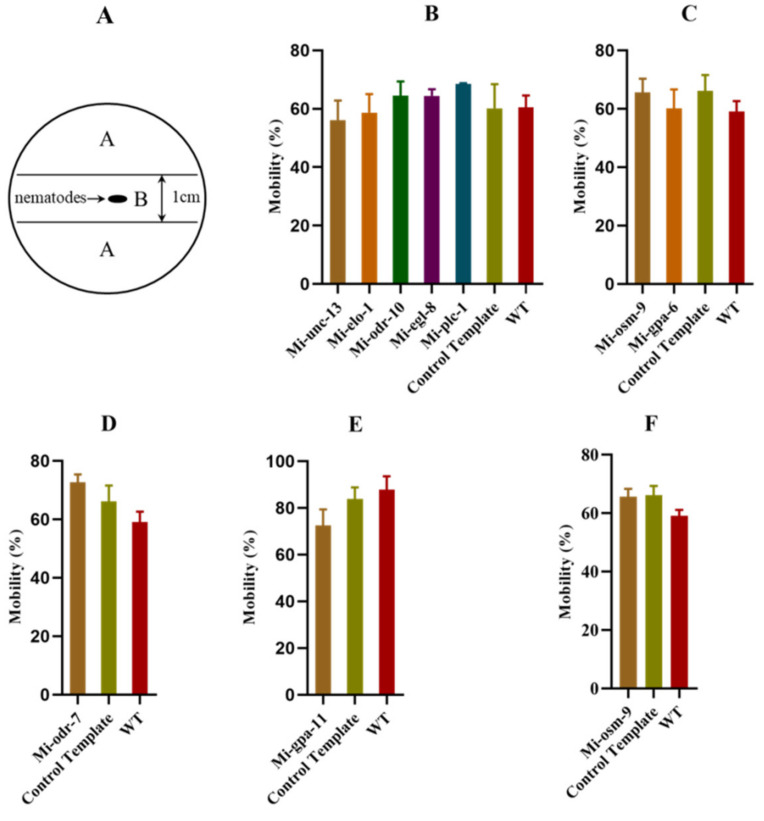
Mobility of *M. incognita* after RNAi with AWA and ASH neurons’ homologous genes: (**A**) the plate was divided into two areas, A and B, to detect the motility of *M. incognita* after RNA interference; (**B**–**D**) the mobility of *M. incognita* after RNAi for the AWA neuronal homologous gene; (**E**,**F**) the mobility of *M. incognita* after RNAi for the ASH neuronal homologous gene. Error bars represent the standard error of the mean.

**Figure 3 pathogens-11-01322-f003:**
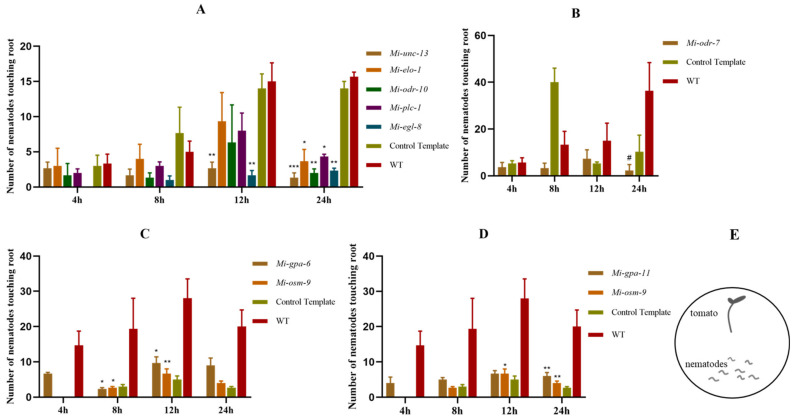
Chemotaxis to tomato roots of *M. incognita* after RNAi with AWA and ASH neurons’ homologous genes. (**A**–**C**) chemotaxis of *M. incognita* after RNAi for the AWA neuronal homologous gene; (**D**) chemotaxis of *M. incognita* after RNAi for the ASH neuronal homologous gene. (**E**) the plate used to test the chemotaxis of *M. incognita*. Error bars represent the standard error of the mean. (Two–way ANOVA; Tukey’s multiple-comparison test; * *p* < 0.05, ** *p* < 0.01, *** *p* < 0.001 and # *p* < 0.0001.).

**Figure 4 pathogens-11-01322-f004:**
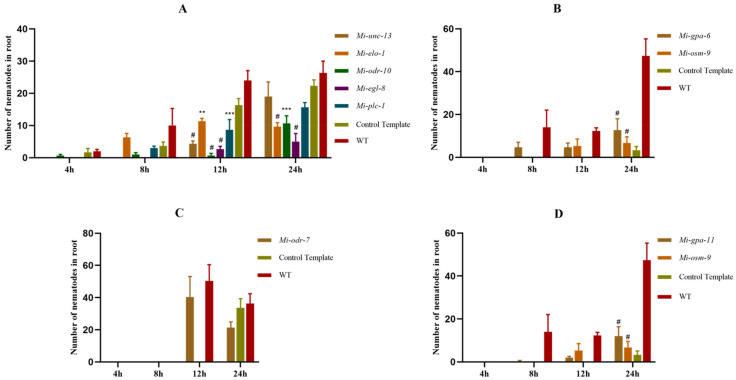
Infection of *M. incognita* on tomato roots after RNA interference with AWA and ASH homologous genes. (**A**–**C**) the infection of *M. incognita* after RNAi for the AWA neuronal homologous gene; (**D**) the infection of *M. incognita* after RNAi for the ASH neuronal homologous gene. Error bars represent the standard error of the mean. (Two–way ANOVA; Tukey’s multiple-comparison test; ** *p* < 0.01, *** *p* < 0.001 and # *p* < 0.0001.).

**Figure 5 pathogens-11-01322-f005:**
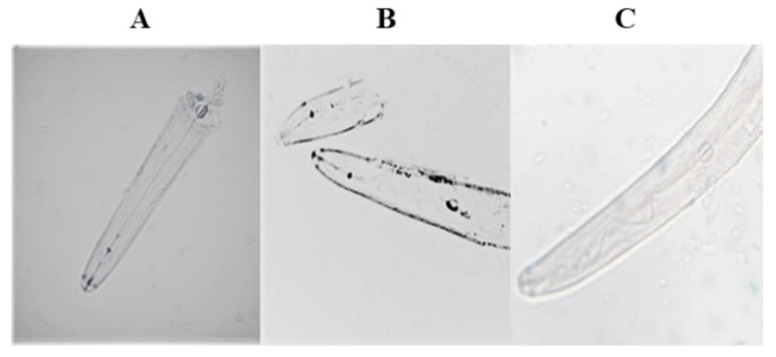
In situ hybridization for homologous genes of *M. incognita*: (**A**) the in situ hybridization of *Mi-gpa-6*; (**B**) the in situ hybridization of *Mi-odr-10*; (**C**) the in situ hybridization of the wild type.

## Data Availability

Not applicable.

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
