# Peer review of "AWA and ASH Homologous Sensing Genes of Meloidogyne incognita Contribute to the Tomato Infection Process"

_pathogens, 2022, doi:10.3390/pathogens11111322_

Round 1
Reviewer 1 Report (Previous Reviewer 1)
Authors have addressed properly all the questions ans suggestions given by the referee.
Just a minor question in page 2, line 21. Delete the word "instar". There are no "instar larvae" in nematodes like in arthropods, just larvae or juveniles.
There are still some typo and punctuation mistakes (missing spaces between some words or after full dots) in the text that should be corrected before the manuscript is published.
Author Response
Ref: Submission ID pathogens-2010120
Responds for reviewers
Reviewer 1
Authors have addressed properly all the questions ans suggestions given by the referee.
Just a minor question in page 2, line 21. Delete the word "instar". There are no "instar larvae" in nematodes like in arthropods, just larvae or juveniles.
There are still some typo and punctuation mistakes (missing spaces between some words or after full dots) in the text that should be corrected before the manuscript is published.
It has been revised.
Please refer to the attachment for specific modifications.

Reviewer 2 Report (New Reviewer)
I have not identified any flows of the methodological approach, the results and the conclusions. It is an interesting paper that deserves to be published prior to some careful reading to omit some minor linguistic/grammatical mistakes.
I only have one minor concern, in the material and methods section, paragraph 4.1 a reference is needed. Some could be: https://link.springer.com/article/10.1007/s10658-021-02360-2, https://doi.org/10.3390/agronomy10081119
Author Response
Ref: Submission ID pathogens-2010120
Responds for reviewers
Reviewer 2
I have not identified any flows of the methodological approach, the results and the conclusions. It is an interesting paper that deserves to be published prior to some careful reading to omit some minor linguistic/grammatical mistakes.
I only have one minor concern, in the material and methods section, paragraph 4.1 a reference is needed. Some could be: https://link.springer.com/article/10.1007/s10658-021-02360-2, https://doi.org/10.3390/agronomy10081119
It has been revised. Relevant literature has been supplemented in the paragraph 4.1.
Please refer to the attachment for specific modifications.

This manuscript is a resubmission of an earlier submission. The following is a list of the peer review reports and author responses from that submission.
Round 1
Reviewer 1 Report
The manuscripts report on the perception of tomato root signals by Meloidogyne incognita and relates homologous genes to those in the C. elegans AWA and ASH neuronal perception pathways for sensing tomato roots. The study is clearly explained, and results are of interesting in elucidating the process of the plant host perception by M. incognita.
It seems that authors are not aware of the studies of Shivakumara et al. 2019. MPMI 32,7: 876-887 (https://doi.org/10.1094/MPMI-08-18-0226-R). They used the knowledge on C. elegans chemosensation to identify four genes in the parasitic nematode Meloidogyne incognita, that are induced in infectious larvae and expressed in chemosensory neurons. These authors showed in knockdown experiments that these genes are required for chemotaxis towards root exudate and for infection of the plant host. These studies are very similar to those shown in this manuscript, and therefore deserve to be included in the discussion. Authors should focus in the advance on the knowledge given by their study in comparison to the results given by Shivakumara et al. Otherwise, the originality of this study may be compromised.
Other comments and suggestions:
Abstract
Page 1, line 15. “…for sensing tomato roots…” change to “…for sensing chemical signals from tomato roots…”
Introduction
Page 2, line 52. “Zhou et al. found…” There is no reference for Zhou et al. Please include it.
Page 2, line 63: “Bargmann et al…”. But the reference number 21 address to a reference in which the first author is Sengupta. Please correct the text accordingly.
Page 2, line 78. “…thetomato…” Please add space between “the” and “tomato”.
Results
Page 3, line 96. “…SEMs…” Please indicate the meaning of SEM, in the text or spell out in the figure legend. Is it the standard error of the mean?
Page 5, figure 4. The quality of the pictures is low. If possible, it should be improved.
Discussion
Page 6, line 194. “Lans determined …” But the reference number 29 address to a reference in which the first author is Silva. Please correct the text accordingly.
Page 6, line 195. “Nagendrappa et al…” But the reference number 30 address to a reference in which the first author is Zhang. Please correct the text accordingly.
Page 6, line 197. “…Mi-tax-4and…” Please separate the word “and” by space.
References
Page 9, line 351. The title in this reference is in capital letters, please change it to small letters.
Reduced infected roots may be just a consequence of the reduced ability to find the roots. This should be included in the discussion.
Author Response
参考:提交 ID 病原体-1817276
回复审稿人
抽象的
第 1 页,第 15 行。“……用于检测番茄根部……”更改为“……用于检测番茄根部的化学信号……”
它已经改变了。
介绍
第 2 页,第 52 行。找到了……”周等人没有参考资料。请包括它。
它已经改变了。引言中增加了一些关于 AWA 神经元的新表达(ex 可被 AWA 和 AWC 神经元检测到的挥发性化学物质主要包括醇类、酮类、酯类、胺类、有机酸类、芳烃类和杂环类,主要由细菌代谢产生。删除秀丽隐杆线虫AWA受体基因odr-10的缺失导致对挥发性化合物二乙酰的感知缺陷,AWA神经元需要odr-7来感知大部分化合物,而odr-7的缺失导致AWA神经元嗅觉受损感觉。
第 2 页,第 63 行:“Bargmann 等人……”。但是参考编号 21 指向第一作者是 Sengupta 的参考文献。请相应地更正文本。
它已被修改。
第 2 页,第 78 行。“...thetomato...” 请在“the”和“tomato”之间添加空格。
它已经改变了。
结果
第 3 页,第 96 行。“……SEMs……” 请在正文中注明 SEM 的含义或在图例中拼出。它是平均值的标准误吗?
SEM 的含义已在文章中说明。SEM 代表平均值的标准误差。
第 5 页,图 4。图片质量低。如果可能的话,应该改进它。
它已被调整。
讨论
第 6 页,第 194 行。“Lans 确定……”但参考编号 29 指向第一作者是 Silva 的参考。请相应地更正文本。
它已被调整。
第 6 页,第 195 行。“Nagendrappa 等人……”但参考编号 30 指向第一作者是张的参考文献。请相应地更正文本。
它已被调整。
第 6 页,第 197 行。“...Mi-tax-4and...” 请用空格分隔“and”一词。
它已被调整。
参考
第 9 页,第 351 行。本参考文献中的标题为大写字母,请改为小写字母。
它已被调整。
受感染的根减少可能只是寻找根的能力降低的结果。这应该包括在讨论中。
这些手稿报告了 南方根结线虫对番茄根信号的感知, 并将同源基因与 线虫 AWA 和 ASH 感知番茄根的神经元感知途径中的基因相关联。该研究解释清楚,结果对阐明 M. incognita对植物宿主感知的过程很感兴趣。
似乎作者不知道 Shivakumara 等人的研究。2019. MPMI 32,7: 876-887 ( https://doi.org/10.1094/MPMI-08-18-0226-R )。
他们利用关于 秀丽隐杆线虫 化学感应的知识来鉴定寄生线虫 南方根结线虫中的四个基因,这些基因在感染性幼虫中被诱导并在化学感觉神经元中表达。这些作者在敲低实验中表明,这些基因对于根系分泌物的趋化性和植物宿主的感染是必需的。这些研究与本手稿中的研究非常相似,因此值得讨论。与 Shivakumara 等人给出的结果相比,作者应该提前关注他们的研究提供的知识。否则,本研究的原创性可能会受到影响。
对引言和讨论进行了修订,结合 Shivakumara 等人的相关内容对我们的研究进行了讨论。2019. MPMI 32,7: 876-887 (https://doi.org/10.1094/MPMI-08-18-0226-R)。
有关详细信息,请参阅附件中上传的 PDF。

Reviewer 2 Report
The manuscript entitled " AWA and ASH homologous sensing genes of Meloidogyne incognita contribute to the tomato infection process” has been submitted by Li et al. Authors explored the role of sensing genes in the infection process of the plant parasitic nematode Meloidogyne incognita. Such approaches have already been performed, even with different candidate genes (for ex, Shivakumara et al, MPMI, 2019), but they are not cited in the paper.
My main remark concerns the absence of control in RNAi experiments. Authors must use dsRNA against Mi unrelated sequences to show that the RNAi treatment itself does not affect the motility of the nematode.
Moreover, the manuscript is not clear and must be carefully rewritten. Some results must be described (description of genes of interest, QPCR data,…). Confusion between Ce and Mi genes occurred all along the manuscript (and between genes and proteins). The methods must be described carefully. Even the legends of figures are not adequate (ex fig4!).
Author Response
Ref: Submission ID pathogens-1817276
Responds for reviewers
Reviewer 2
The manuscript entitled " AWA and ASH homologous sensing genes of Meloidogyne incognita contribute to the tomato infection process” has been submitted by Li et al. Authors explored the role of sensing genes in the infection process of the plant parasitic nematode Meloidogyne incognita. Such approaches have already been performed, even with different candidate genes (for ex, Shivakumara et al, MPMI, 2019), but they are not cited in the paper.
The introduction and discussion were revised, and our research was discussed in combination with the relevant content of Shivakumara et al. 2019. MPMI 32,7: 876-887 (https://doi.org/10.1094/MPMI-08-18-0226-R).
My main remark concerns the absence of control in RNAi experiments. Authors must use dsRNA against Mi unrelated sequences to show that the RNAi treatment itself does not affect the motility of the nematode.
In the experiment we added, we added a sequence unrelated to the Meloidogyne incognita dsRNA as a control template. The control template for RNAi experiment of non mammalian system provided by MEGAscript RNAi kit(AM1626,Thermo Fisher Scientific), which can produce a 500 bp dsRNA product.
Moreover, the manuscript is not clear and must be carefully rewritten. Some results must be described (description of genes of interest, QPCR data,…). Confusion between Ce and Mi genes occurred all along the manuscript (and between genes and proteins). The methods must be described carefully. Even the legends of figures are not adequate (ex fig4!).
We have carefully revised the content in the manuscript, described the relevant genes, and added the QPCR experimental data.
The gene of Caenorhabditis elegans is expressed as osm-9, the related protein is expressed as OSM-9, and the gene of Meloidogyne incognita is expressed as Mi-osm-9. The legend has been changed.
